# Kaffir Lime Essential Oil Variation in the Last Fifty Years: A Meta-Analysis of Plant Origins, Plant Parts and Extraction Methods

Rahmat Budiarto [1,2,*] and Mohammad Miftakhus Sholikin [3,4]

1   Department of Agronomy, Faculty of Agriculture, Universitas Padjadjaran, Sumedang 45363, Indonesia
2   Meta-Analysis in Plant Science (MAPS) Research Group, Bandung 40621, Indonesia
3   Research Group of the Technology for Feed Additive and Supplement, Research Center for Animal Husbandry, Research Organization of Agriculture and Food, National Research and Innovation Agency (BRIN), Yogyakarta 55861, Indonesia
4   Animal Feed and Nutrition Modelling Research Group, IPB University, Cibinong 16911, Indonesia
*   Correspondence: rahmat.budiarto@unpad.ac.id

**Abstract:** Kaffir lime has been widely researched for use as an essential oil, alongside its main function as an Asian spice, due to the needs of numerous industries. A meta-analysis was used to summarize the variation in yield and main components of kaffir lime essential oils in response to differences in plant origins, plant parts, and extraction methods during the last fifty years. A database was constructed from 85 data items from 36 single studies, prepared by following PRISMA-P. The result showed no significant effect of extraction method on variation in oil yield and main components. In contrast, after integrating numerous single studies under a statistical approach, some interesting facts emerged, such as (i) plant origin significantly affected the citronellol, i.e., subtropical conditions were more favorable than tropical ones; (ii) caryophyllene was found in all countries, from subtropical America and tropical Asia to tropical East Africa; (iii) the richest parts of citronellal, citronellol, citronellyl acetate, and sabinene were leaves, blossoms, twigs, and fruit peels, respectively; and (iv) due to significant interaction of plant origin and plant parts, a very challenging issue in boosting oil yield could be answered by formulating location-specific and organ-specific culture practices. This study had succeeded in providing further research directions.

**Keywords:** *Citrus hystrix* DC; citronellal; citronellol; citronellyl acetate; caryophyllene; meta-analysis; linalool; PRISMA-P; sabinene; yield

## 1. Introduction

Kaffir lime (*Citrus hystrix* DC) is one of the less popular citrus varieties, due to its unpalatable fruit in contrast to other common citrus species [1,2]. This plant is generally cultured in the yards of local residents by using a polyculture system rather than in intensive monoculture systems, like other species [3]. While earlier studies have revealed the valleys of the Southeastern Himalayas as the original location of common citrus, the kaffir lime is thought to originate from the Southeast Asian region [4,5]. One unique characteristic that is easily found in kaffir lime, rather than common citrus, is the aromatic leaf with an eight-shape, also known as bifoliate leaf [6–8]. The leaf is more popularly used than its fruit, as the leaf can be found in numerous Asian cooking recipes [3,9].

Numerous species of the *Citrus* genus are evidently utilized as essential oils [10], including kaffir lime [11–13]. In general, *Citrus*-originated essential oils show various beneficial pharmacological characteristics, such as antioxidant [14], anticancer [15], antimicrobial [16], antiparasitic [17], antifungal [18], antibacterial [19,20], anti-inflammatory [21], antifeedant [22], biolarvicide [23,24], and sleep relaxation properties [25]. Essential oil is a highly competitive downstream *Citrus* product with bright development prospects for the

future, alongside the complexity and growth of human needs, such as for food additives, perfumery, cosmetics, and pharmaceutical industries. The essential oil of kaffir lime has a good opportunity to further develop due to its strong odor characteristic [1,3].

Kaffir lime essential oil production requires a high number of raw materials support that can be achieved though plant production improvement. Not only extensification in terms of arable land expansion, but also intensification of agricultural inputs is believed to be the main strategy for boosting plant production to meet the market demand [26]. Earlier studies have reported intensification culture techniques to improve kaffir lime production, including the modification of canopy pruning [27], application of artificial shading [13] and fertilizer [28,29], and soil amendments [29]. Although most local farmers use the leaf as the raw material of essential oil, some industries overseas explore the possibility of extracting certain essential oils from the peel of the fruit [3]. The different plant parts may result in different yields and main components of essential oil, as reported by earlier studies [1,30–37]. In addition to plant parts, numerous studies have revealed that both yield and the main components of essential oils are also affected by the plant origin [11,30,38–40] and extraction method [41–47].

Numerous single studies on the effects of plant part, plant origin, and extraction method have been widely published in recent years. However, the results still show a highly variable effect of these factors on oil yield and main components. The discrepancy among the numerous single studies is caused by the limitation of single studies in providing a precise estimate of an effect. In order to have a greater understanding of the magnitude of effect reported by numerous published single studies, irrespective of the place and time of experiment, the present study used a meta-analysis approach. This statistical approach is widely used to produce a weighted average of numerous earlier study results, conceive the resulting pattern, and calculate the uncertainty value of the estimated equation [48,49]. Compared to single studies, the positive aspects of meta-analysis are stronger statistical power, bigger sample size, more efficiency, and greater accuracy to form a comprehensive conclusion. This meta-analysis is more evidence-based, and also differs from the conventional review approach, in which reviewers generally write their own data interpretation without considering quantitative statistics. The meta-analysis approach has been frequently applied to citrus, for example, to summarize water and nitrogen use efficiencies [50], varietal selection [51], Huanglongbing resistance gene identification [52], and the relationship of fruit intake and cancer risk [53–55]. However, there is still limited research specific to kaffir lime essential oils. Therefore, the present study aimed to elucidate the variation in yield and main components of kaffir lime essential oils in response to difference plant parts, plant origin, and extraction method during the last fifty years by using a meta-analysis approach.

## 2. Materials and Methods

### 2.1. Searching

The references used, henceforth called studies, were globally indexed journals and book chapters. The search for studies was carried out from July 2021 to July 2022 by using Google Scholar search engine, with the keyword "kaffir lime metabolite". The search activities resulted in finding 591 studies published in the last fifty years (1970 up to 2022). The selection was carried out in this study by taking into account the relevance of the literature to the research topic, i.e., the yield and the content of the main metabolites, such as citronellal, citronellol, linalool, citronellyl acetate, sabinene, and caryophyllene in kaffir lime essential oil.

### 2.2. Selection

Selection was carried out to ensure that all selected journals were (i) indexed at least by Google Scholar; (ii) equipped with digital object identifier/doi or uniform resource locator/url; and (iii) reported plant origin; plant part, and the plant extraction method used, as well as observation variables, such as the yield and content of the main abovementioned

metabolites, that were presented quantitatively either in table or graph form. Studies with qualitative data were excluded. In total, there were 36 single studies that passed the selection for database construction.

*2.3. Tabulation*

The reference extraction process on 36 obtained studies resulted in 85 data points that were further tabulated in the database, as shown in Table 1 and Table S1 in Supplementary Materials, for further meta-analysis purposes. The meta-analysis procedure in the present research followed the guideline handbook by [56]. Variables extracted from every study were year of publication, plant origin, plant part, doi or url, and several metabolites (citronellal, citronellol, linalool, citronellyl acetate, sabinene, and caryophyllene) content. The meta-analysis study was prepared by following the Preferred Reporting Items for Systematic review and Meta-Analysis Protocols (PRISMA-P) [57]. If there was variation in the observation unit of certain variables, unit conversion was carried out by following the standard of the international system.

**Table 1.** Database for meta-analysis of kaffir lime essential oil variation in the last fifty years.

| No. | Publication Year | Plant Origin | Plant Part | Extraction Method | Ref. |
|-----|------------------|--------------|------------|-------------------|------|
| 1. | 1971 | Thailand | Leaves | SDD | [31] |
| 2. | 1971 | Thailand | Peels | CPE | [31] |
| 3. | 1990 | Chiang Rai, Thailand | Leaves | SDD | [33] |
| 4. | 1990 | Chiang Rai, Thailand | Twigs | SDD | [33] |
| 5. | 1990 | Chiang Rai, Thailand | Peels | SDD | [33] |
| 6. | 1990 | Chiang Rai, Thailand | Peels | SEE | [33] |
| 7. | 1990 | Chiang Rai, Thailand | Fruit | SEE | [33] |
| 8. | 1996 | Melaka, Malaysia | Leaves | HD | [35] |
| 9. | 1996 | Melaka, Malaysia | Peels | HD | [35] |
| 10. | 1999 | Terengganu, Malaysia | Leaves | SDD | [34] |
| 11. | 1999 | Terengganu, Malaysia | Leaves | LNE | [34] |
| 12. | 1999 | Terengganu, Malaysia | Peels | LNE | [34] |
| 13. | 2007 | Thailand, | Peels | HD | [58] |
| 14. | 2007 | Bangkok, Thailand | Leaves | SEE | [46] |
| 15. | 2007 | Bangkok, Thailand | Leaves | SPE | [46] |
| 16. | 2008 | Songkla, Thailand | Peels | SEE | [47] |
| 17. | 2008 | Songkla, Thailand | Peels | HD | [47] |
| 18. | 2009 | Florida, US | Blossoms | SPE | [59] |
| 19. | 2010 | Khon kaen, Thailand | Leaves | HD | [60] |
| 20. | 2010 | Khon kaen, Thailand | Peels | CPE | [60] |
| 21. | 2010 | Chiang Mai, Thailand | Leaves | SEE | [61] |
| 22. | 2010 | Chiang Mai, Thailand | Peels | SEE | [61] |
| 23. | 2010 | Chiang Mai, Thailand | Twigs | SEE | [61] |
| 24. | 2011 | Selangor, Malaysia | Leaves | HD | [62] |
| 25. | 2012 | Bangkok, Thailand | Leaves | SEE | [63] |
| 26. | 2012 | Thailand | Leaves | SPE | [1] |
| 27. | 2012 | Thailand | Peels | SPE | [1] |

**Table 1.** *Cont.*

| No. | Publication Year | Plant Origin | Plant Part | Extraction Method | Ref. |
|-----|-----|-----|-----|-----|-----|
| 28. | 2012 | Thailand | Leaves | SDD | [64] |
| 29. | 2013 | Selangor, Malaysia | Peels | SDD | [45] |
| 30. | 2013 | Selangor, Malaysia | Peels | SDD | [45] |
| 31. | 2013 | Selangor, Malaysia | Peels | SDD | [45] |
| 32. | 2013 | Selangor, Malaysia | Peels | SDD | [45] |
| 33. | 2013 | Selangor, Malaysia | Peels | HDSD | [44] |
| 34. | 2013 | Selangor, Malaysia | Peels | HDSD | [44] |
| 35. | 2013 | Selangor, Malaysia | Peels | HDSD | [44] |
| 36. | 2013 | Selangor, Malaysia | Peels | HDSD | [44] |
| 37. | 2013 | Selangor, Malaysia | Peels | SDD | [43] |
| 38. | 2013 | Selangor, Malaysia | Peels | SDD | [43] |
| 39. | 2013 | Selangor, Malaysia | Peels | SDD | [43] |
| 40. | 2013 | Chiang Mai, Thailand | Leaves | SFE | [65] |
| 41. | 2013 | Chiang Mai, Thailand | Leaves | SFE | [65] |
| 42. | 2013 | Chiang Mai, Thailand | Leaves | SFE | [65] |
| 43. | 2013 | Chiang Mai, Thailand | Leaves | SFE | [65] |
| 44. | 2013 | Chiang Mai, Thailand | Leaves | SFE | [65] |
| 45. | 2013 | Chiang Mai, Thailand | Leaves | SFE | [65] |
| 46. | 2013 | Chiang Mai, Thailand | Leaves | SFE | [65] |
| 47. | 2013 | Chiang Mai, Thailand | Leaves | SFE | [65] |
| 48. | 2013 | Chiang Mai, Thailand | Leaves | SFE | [65] |
| 49. | 2013 | Chiang Mai, Thailand | Leaves | SFE | [65] |
| 50. | 2014 | Thai-Cina/Thailand | Leaves | - | [66] |
| 51. | 2016 | Réduit, Mauritius | Peels | HD | [67] |
| 52. | 2017 | Tulungagung, Indonesia | Twigs and leaves | SDD | [30] |
| 53. | 2017 | Tulungagung, Indonesia | Twigs | SDD | [68] |
| 54. | 2017 | Blitar, Indonesia | Leaves | HSD | [68] |
| 55. | 2017 | Blitar, Indonesia | Peels | HSD | [68] |
| 56. | 2017 | Malang, Indonesia | Leaves | SEE | [69] |
| 57. | 2018 | Pallepola, Sri Lanka | Peels | SPE | [70] |
| 58. | 2018 | Lampang, Thailand | Leaves | SEE | [71] |
| 59. | 2018 | Chiang Mai, Thailand | Peels | HD | [72] |
| 60. | 2019 | Thai-Cina/Thailand | Peels | - | [32] |
| 61. | 2019 | Thai-Cina/Thailand | Leaves | - | [32] |
| 62. | 2019 | Klaten, Indonesia | Leaves | HD | [42] |
| 63. | 2019 | Klaten, Indonesia | Dry leaves (1 dD) | HD | [42] |
| 64. | 2019 | Klaten, Indonesia | Dry leaves (2 dD) | HD | [42] |
| 65. | 2019 | Klaten, Indonesia | Dry leaves (3 dD) | HD | [42] |
| 66. | 2019 | Klaten, Indonesia | Dry leaves (4 dD) | HD | [42] |
| 67. | 2019 | Klaten, Indonesia | Dry leaves (5 dD) | HD | [42] |

**Table 1.** *Cont.*

| No. | Publication Year | Plant Origin | Plant Part | Extraction Method | Ref. |
|---|---|---|---|---|---|
| 68. | 2019 | Klaten, Indonesia | Dry leaves (6 dD) | HD | [42] |
| 69. | 2019 | Klaten, Indonesia | Dry leaves (7 dD) | HD | [42] |
| 70. | 2019 | Klaten, Indonesia | Dry leaves (8 dD) | HD | [42] |
| 71. | 2019 | An Giang, Vietnam | Peels | SDD | [73] |
| 72. | 2020 | An Giang, Vietnam | Peels | HD | [74] |
| 73. | 2020 | An Giang, Vietnam | Peels | SFME | [75] |
| 74. | 2020 | Bangkok, Thailand | Peels | SEE | [37] |
| 75. | 2020 | Bangkok, Thailand | Leaves | SEE | [37] |
| 76. | 2021 | Tulungagung, Indonesia | Leaves | HSD | [11] |
| 77. | 2021 | Bogor, Indonesia | Leaves | HSD | [11] |
| 78. | 2021 | Pasuruan, Indonesia | Leaves | HSD | [11] |
| 79. | 2021 | West Bandung, Indonesia | Leaves | HSD | [11] |
| 80. | 2021 | An Giang, Vietnam | Peels | SDD | [76] |
| 81. | 2021 | An Giang, Vietnam | Peels | HD | [77] |
| 82. | 2022 | Central Java, Indonesia | Fresh leaves | HD | [78] |
| 83. | 2022 | East Sumba, Indonesia | Fresh leaves | HD | [78] |
| 84. | 2022 | Central Java, Indonesia | Peels | HD | [78] |
| 85. | 2022 | East Sumba, Indonesia | Peels | HD | [78] |

Ref.—references, CPE—cold press extraction, HDSD—hydro diffusion steam distillation, HD—hydro distillation, HSD—hydro steam distillation, LNE—Likens-Nikerson extraction, SPE—solid phase extraction, SEE—solvent extraction, SFME—solvent free microwave extraction, SDD—steam distillation, SFE—supercritical fluid extraction, dD—day of drying.

*2.4. Modelling*

The method of meta-analysis used in the present experiment refers to the linear mixed model (LMM) [48,49]. The mathematical models are shown below.

$$Y_{ij} = \mu + s_i + \tau_j + s\tau_{ij} + \beta_0 + \beta_1 P_{ij} + b_i P_{ij} + e_{ij} \tag{1}$$

$$Interaction = PS + O + EM + PS * O + PS * EM + O * EM + PS * O * EM \tag{2}$$

Note: $Y_{ij}$ is a dependent variable, $\mu$ is the overall mean value, $s_i$ is the $i$-th random factor of the study difference, $\tau_j$ is the $j$-th fixed factor of the predictor, $s\tau_{ij}$ is a factor of random interaction between the difference of the study and the fixed factor of the predictor, $\beta_0$ is the value of the intersection of the average of all studies with the axis, $\beta_1$ is the regression coefficient, $P_{ij}$ is the amount of predictor, $b_i$ is the random effect of the difference in studies from the regression coefficient Y in the X of the $i$-th study, $e_{ij}$ is the unexplained error value, *EM* is the extraction method, *O* is the origin, and *PS* is part of the sample. Meta-regression followed Equation (1). The advanced stage of the meta-analysis test was the least-squares means advanced test [79]. Interaction effects were sought by using Equation (2).

*2.5. Determination of Numeric Predictors and Their n-Encoders*

There were three categories of quantitative predictors used, namely, plant origin, plant part for sample, and plant extraction method. The mentioned three predictors were analyzed separately, and each component was encapsulated in the form of ordinal data in alphabetical order. Plant origin predictor components were (1) Indonesia, (2) Malaysia, (3) Mauritius, (4) Sri Lanka, (5) Thailand, (6) USA, and (7) Vietnam. The predictors of

plant part for a sample were encoded as follows: blossom/B (1), dry leaves/DL (2), fresh leaves/FL (3), fruits/F (4), leaves/L (5), peels/P (6), and twigs/T (7). Plant extraction method predictors included cold press extraction/CPE (1), hydro diffusion steam distillation/HDSD (2), hydro distillation/HD (3), hydro steam distillation/HSD (4), Likens-Nikerson extraction/LNE (5), solid phase extraction/SPE (6), solvent extraction/SEE (7), solvent free microwave extraction/SFME (8), steam distillation/SDD (9), and supercritical fluid extraction/SFE (10).

*2.6. Statistical Test of Model*

Statistical analysis was performed by using R version 4.2.0 and statistical assay, using residual mean square error (RMSE) and Nakagawa determination coefficient or $R_{GLMM}(c)^2$ [80–82]. The equations *RMSE* (3) and $R^2$ Nakagawa (4) are as follows:

$$RMSE = \sqrt{\frac{\sum(O-P)^2}{NDP}} \tag{3}$$

$$R_{GLMM}(c)^2 = \frac{\left(\sigma^2_f + \sum(\sigma^2_l)\right)}{\left(\sigma^2_f + \sum(\sigma^2_l) + \sigma^2_e + \sigma^2_d\right)} \tag{4}$$

where $O$ = observation value (actual value), $P$ = predicted value, $NDP$ = number of data point, $\sigma^2_f$ is the variant of a fixed factor, $\sum(\sigma^2_l)$ is the sum of all variants of the component, $\sigma^2_e$ is the variant due to the predictor dispersion, and $\sigma^2_d$ is the specific distribution of the variant. Then, to measure the significance of the model, a variance analysis test was carried out, which is significant if $p < 0.05$ and tended to be significant if $p < 0.1$.

## 3. Results

*3.1. Descriptive Statistics*

Descriptive statistics of the tabulated database in the present meta-analysis are depicted in Table 2. The number of data points (NDP) in all assessed variables was higher than 10, implying that minimum eligibility had been reached for use as a database for meta-analysis, following previous studies [83–85]. The maximum, average, and minimum extracted yields from kaffir lime ever recorded were 17.5%, 1.94%, and 0.03%, respectively (Table 2). Variations in yield in response to plant origin, plant part, and extraction method were found to be in the range of 0.43–3.13%, 0.03–10.1%, and 0.19–5.21%, respectively (Table 3). In terms of composition in kaffir lime essential oil, citronellal was the most dominant component according to 71 earlier cases, irrespective of origin, plant part, and extraction method. The highest citronellal content ever recorded was 87.6%, while the lowest result was 0.36% (Table 2). Plant origin, plant part, and extraction method caused a wide range of variation in terms of citronellal, i.e., 6.05–60.1%, 6.05–68.9%, and 1.62–53%, respectively (Table 3). Sabinene was placed precisely under the citronellal, with a maximum proportion of about 48.5%, while the lowest one was 0.2% (Table 2). Sabinene variation was observed in ranges of 5.28–31.4%, 1.71–24.3%, and 4.37–36.7% in response to plant origin, plant part, and extraction method, respectively (Table 3). The other four phytochemicals frequently found to compose the kaffir lime essential oil profile were citronellol, citronellyl acetate, linalool, and caryophyllene, with an average of about 4.45%, 2.8%, 4.29%, and 1.12%, respectively (Table 2).

**Table 2.** Data aggregation and descriptive statistics of tabulated database.

| No. | Parameters | NDP | Mean | SD | Max | Min | Q25 | Q50 | Q75 |
|---|---|---|---|---|---|---|---|---|---|
| | Main components of kaffir lime essential oils, % of total essential oil | | | | | | | | |
| 1. | Caryophyllene | 46 | 1.12 | 1.21 | 3.99 | 0 | 0.32 | 0.6 | 1.34 |
| 2. | Citronellal | 71 | 36.6 | 31.7 | 87.6 | 0.36 | 7.93 | 20.9 | 70.4 |
| 3. | Citronellol | 53 | 4.45 | 4.85 | 25.3 | 0.1 | 1.32 | 2.91 | 6.35 |
| 4. | Citronellyl acetate | 28 | 2.8 | 2.39 | 7.78 | 0.12 | 0.48 | 1.96 | 4.31 |
| 5. | Linalool | 56 | 4.29 | 11.4 | 86.1 | 0.03 | 0.89 | 1.89 | 3.89 |
| 6. | Sabinene | 46 | 15.5 | 16.4 | 48.5 | 0.2 | 1.82 | 5.91 | 23.5 |
| | Yield of kaffir lime essential oils, % from fresh weight | | | | | | | | |
| 7. | Extraction yield | 55 | 1.94 | 2.69 | 17.5 | 0.03 | 0.52 | 1.26 | 2.19 |

Max—Maximum value of the feature data, Min—minimum value of the feature data, NDP—Number of data points, SD—Standard deviation, Q25—Quantile data 25%, Q50—Quantile data 50%, Q75—Quantile data 75%.

**Table 3.** Mean ($\overline{X}$) and standard deviation (SD) based on plant origin, plant part, and extraction method.

| No. | Predictor | Caryophyllene | | Citronellal | | Citronellol | | Citronellyl Acetate | | Linalool | | Sabinene | | Yield | |
|---|---|---|---|---|---|---|---|---|---|---|---|---|---|---|---|
| | | $\overline{X}$ | SD | $\overline{X}$ | SD | $\overline{X}$ | SD | $\overline{X}$ | SD | $\overline{X}$ | SD | $\overline{X}$ | SD | $\overline{X}$ | SD |
| | Plant origin | | | | | | | | | | | | | | |
| 1. | Indonesia | 0.97 | 0.83 | 60.1 | 24.6 | 7.98 | 4.15 | 4.51 | 2.63 | 10.4 | 21.3 | 5.28 | 7.1 | 1.05 | 0.34 |
| 2. | Malaysia | 0.48 | 0.28 | 22.9 | 26.3 | 3.9 | 4.38 | 1.38 | 1.39 | 1.2 | 0.88 | 31.4 | 17.5 | 2.84 | 1.6 |
| 3. | Mauritius | 0.13 | | | | | | | | | | | | 0.43 | |
| 4. | Sri Lanka | 3.99 | | 12.3 | | 1.32 | | | | 4.02 | | | | | |
| 5. | Thailand | 1.27 | 1.31 | 31.3 | 32.7 | 3.14 | 3 | 3.15 | 2.42 | 2.83 | 1.67 | 7.61 | 8.6 | 1.75 | 3.97 |
| 6. | USA | 3.75 | | 6.05 | | 25.3 | | | | 4.86 | | 18.1 | | | |
| 7. | Vietnam | 0.45 | 0.04 | 13.5 | 3.48 | 1.7 | 1.27 | 0.27 | | 0.83 | 0.25 | 17.9 | 6.99 | 3.13 | 1.5 |
| | Plant part | | | | | | | | | | | | | | |
| 8. | Blossom | 3.75 | | 6.05 | | 25.3 | | | | 4.86 | | 18.1 | | | |
| 9. | Dry leaves | | | 68.9 | 13.5 | | | | | | | | | 1 | 0.19 |
| 10. | Fresh leaves | 0.26 | | 57 | | 11.7 | | 1.74 | | 46.5 | | 56 | 1.79 | | |
| 11. | Fruit | 0.83 | | | | | | 0.45 | | 0.81 | | 4.52 | | 0.03 | |
| 12. | Leaves | 1.37 | 1.28 | 53.3 | 32.3 | 5.44 | 4.15 | 3.74 | 2.22 | 3.08 | 1.11 | 1.71 | 1.26 | 1.06 | 1.67 |
| 13. | Peel | 0.86 | 1.12 | 10.7 | 6.34 | 1.67 | 1.35 | 1.46 | 1.77 | 1.64 | 1.7 | 24.3 | 16.1 | 10.1 | 1.62 |
| 14. | Twigs | 1.04 | 0.76 | 57.2 | 18.6 | 9.47 | 2.7 | 5.81 | 1.65 | 10.9 | 3.88 | 4.01 | 3.29 | 1.04 | 1.32 |
| | Extraction method | | | | | | | | | | | | | | |
| 15. | Cold press extraction | 2.02 | 2.43 | 14 | 13.9 | 0.94 | 0.76 | 2.01 | 2.56 | 2.36 | 2.63 | 12.1 | 14.9 | | |
| 16. | Hydro diffusion steam distillation | 0.4 | 0.22 | 11.6 | 4.09 | 1.26 | 0.66 | | | 0.77 | 0.16 | 36.7 | 7.4 | 2.24 | 1.1 |
| 17. | Hydro distillation | 0.88 | 1.01 | 46.2 | 28 | 4.64 | 4.1 | 2.44 | 2.51 | 10.1 | 23 | 7.23 | 8.82 | 1.83 | 1.94 |
| 18. | Hydro steam distillation | 1.01 | 1.08 | 53 | 45.4 | | | 2.77 | | 3.85 | 0.54 | 6 | 4.54 | | |
| 19. | Likens-Nikerson extraction | | | 42.5 | 42.4 | 6.84 | 4.95 | 1.45 | 0.32 | 1.69 | 0.18 | 11.1 | 12.8 | 1.9 | 0.14 |
| 20. | Solid phase extraction | 2.2 | 1.93 | 15.6 | 18.5 | 9.38 | 10.4 | 5.51 | 3.21 | 4.37 | 0.77 | 8.02 | 9.06 | | |

**Table 3.** *Cont.*

| No. | Predictor | Caryophyllene | | Citronellal | | Citronellol | | Citronellyl Acetate | | Linalool | | Sabinene | | Yield | |
|---|---|---|---|---|---|---|---|---|---|---|---|---|---|---|---|
| | | $\overline{X}$ | SD | $\overline{X}$ | SD | $\overline{X}$ | SD | $\overline{X}$ | SD | $\overline{X}$ | SD | $\overline{X}$ | SD | $\overline{X}$ | SD |
| | | | | | | | Plant origin | | | | | | | | | |
| 21. | Solvent extraction | 2.18 | 1.39 | 40.4 | 39.2 | 4.43 | 4.08 | 2.06 | 2.33 | 2.38 | 1.17 | 4.37 | 5.29 | 5.21 | 6.52 |
| 22. | Solvent free microwave extraction | 0.4 | | 17.8 | | 1.24 | | | | 0.92 | | | | | |
| 23. | Steam distillation | 0.71 | 0.78 | 39.2 | 32.63 | 4.81 | 4.36 | 2.57 | 2.43 | 2.73 | 3.09 | 25.4 | 18.9 | 1.86 | 1.18 |
| 24. | Supercritical fluid extraction | 0.48 | 0.18 | 1.62 | 0.29 | 2.17 | 1.22 | | | | | | | 0.19 | 0.09 |

### 3.2. Meta-Regression on Plant Origin, Plant Part and Extraction Method

The present study reveals the plant origin, plant part, and extraction method effects on yield and main components of kaffir lime essential oil using a quantitative meta-analysis approach. Concerning plant origins, quantitative meta-regression that was carried out on 85 data points revealed that there was no significant effect of plant origin on most variables, namely, yield ($p = 0.482$), citronellol ($p = 0.177$), sabinene ($p = 0.695$), citronellyl acetate ($p = 0.114$), linalool ($p = 0.898$), and caryophyllene ($p = 0.538$). In contrast, the citronellal was significantly influenced by plant origin factor ($p = 0.038$), and it has a negative gradient value forming a decreasing linear pattern (Table 4). In terms of plant sampling part predictor, quantitative meta-regression showed a significant effect only in three parameters, namely, citronellal ($p = 0.01$), citronellol ($p = 0.047$), and citronellyl acetate ($p = <0.001$), with a negative gradient value forming a decreasing linear pattern (Table 4). Concerning extraction method, no parameter was found to be significantly influenced; however, the content of linalool tended to be significantly influenced with a positive gradient value forming an increasing linear regression pattern (Table 4).

**Table 4.** Meta-regression of yield and main components of kaffir lime essential oil in response to plant origin, plant part of sample, and extraction method.

| No. | Predictor | Parameters | Intercepted at Y Axis | | Gradient | | | RMSE | $R^2$ |
|---|---|---|---|---|---|---|---|---|---|
| | | | Value | SE | Value | SE | *p*-Value | | |
| 1. | Plant origin | Caryophyllene | 1.8 | 1.12 | 0.175 | 0.279 | 0.538 | 1.75 | 0.49 |
| 2. | Plant origin | Citronellal | 52.2 | 8.97 | −4.72 | 2.19 | 0.038 | 19.9 | 0.45 |
| 3. | Plant origin | Citronellol | 7.67 | 2.93 | −1.06 | 0.77 | 0.177 | 10.9 | 0.06 |
| 4. | Plant origin | Citronellyl acetate | 7.86 | 2.01 | −0.76 | 0.467 | 0.114 | 1.93 | 0.76 |
| 5. | Plant origin | Linalool | 3.14 | 1.25 | −0.039 | 0.299 | 0.898 | 1.68 | 0.35 |
| 6. | Plant origin | Sabinene | 14.7 | 5.93 | −0.576 | 1.45 | 0.695 | 5.46 | 0.77 |
| 7. | Plant origin | Yield | 0.91 | 0.58 | 0.089 | 0.124 | 0.482 | 0.54 | 0.71 |
| 8. | Plant part | Caryophyllene | 0.74 | 1.89 | 0.301 | 0.329 | 0.365 | 1.79 | 0.45 |
| 9. | Plant part | Citronellal | 77.9 | 16.6 | −7.96 | 2.98 | 0.01 | 18.7 | 0.52 |
| 10. | Plant part | Citronellol | 19.6 | 7.71 | −2.78 | 1.37 | 0.047 | 10.8 | 0.08 |
| 11. | Plant part | Citronellyl acetate | 16.7 | 3.10 | −2.15 | 0.544 | <0.001 | 1.72 | 0.79 |
| 12. | Plant part | Linalool | 0.28 | 2.22 | 0.481 | 0.383 | 0.223 | 1.67 | 0.35 |
| 13. | Plant part | Sabinene | 5.72 | 8.00 | 1.26 | 1.35 | 0.357 | 5.43 | 0.76 |
| 14. | Plant part | Yield | 2.55 | 0.78 | −0.232 | 0.134 | 0.092 | 0.53 | 0.7 |

**Table 4.** *Cont.*

| No. | Predictor | Parameters | Intercepted at Y Axis | | Gradient | | | RMSE | R$^2$ |
|---|---|---|---|---|---|---|---|---|---|
| | | | Value | SE | Value | SE | *p*-Value | | |
| 15. | Extraction method | Caryophyllene | 2.75 | 1.25 | −0.056 | 0.18 | 0.757 | 1.76 | 0.49 |
| 16. | Extraction method | Citronellal | 22.6 | 10.6 | 1.8 | 1.57 | 0.255 | 18.7 | 0.5 |
| 17. | Extraction method | Citronellol | 7.67 | 3.81 | −0.567 | 0.589 | 0.342 | 11.3 | 0.04 |
| 18. | Extraction method | Citronellyl acetate | 2.95 | 2.06 | 0.371 | 0.316 | 0.246 | 1.82 | 0.8 |
| 19. | Extraction method | Linalool | 0.95 | 1.07 | 0.336 | 0.165 | 0.055 | 1.22 | 0.65 |
| 20. | Extraction method | Sabinene | 17.8 | 4.95 | −0.862 | 0.684 | 0.216 | 5.05 | 0.81 |
| 21. | Extraction method | Yield | 1.2 | 0.48 | 0.018 | 0.072 | 0.808 | 0.55 | 0.7 |

R$^2$—R squared, RMSE—Root mean square error, SE—Standard error.

### 3.3. Meta-Analysis on Plant Origin, Plant Part, and Extraction Method

Concerning plant origin, the findings of the meta-analysis showed that the effect of plant origin tends to be significant (*p* = 0.06) on citronellal, with the highest result found in Indonesia (57.1%), while the lowest one was from the USA (6.05%). The order of citronellal content from high to low, respective to plant origin, was Indonesia > Thailand > Malaysia > Vietnam > Sri Lanka > USA. In addition, it was also reported that there was a significant effect of plant origin on the citronellol (*p* < 0.001), sabinene (*p* = 0.029), and caryophyllene (*p* = 0.06). The highest citronellol content was found in kaffir lime from the USA, with Indonesia placing second, while the lowest result was from Sri Lanka. The highest sabinene content was found in kaffir lime samples from Malaysia, which were significantly different from samples from Indonesia, which represented the lowest result. The highest caryophyllene content was found in kaffir lime samples from Sri Lanka, whereas the sample from Mauritius was the lowest result (Table 5).

**Table 5.** Meta-analysis of yield and main components of kaffir lime essential oil in response to plant origin (mean ± standard error).

| No. | Variables | *p*-Val. | Indonesia | Malaysia | Mauritius | Sri Lanka | Thailand | USA | Vietnam |
|---|---|---|---|---|---|---|---|---|---|
| 1. | Caryophyllene | 0.044 | 0.88 ± 0.48 | 0.52 ± 0.7 | 0.13 ± 1.14 | 3.99 ± 1.14 | 1.51 ± 0.31 | 3.75 ± 1.14 | 0.45 ± 0.57 |
| 2. | Citronellal | 0.06 | 57.1 ± 8.42 | 26.1 ± 8.95 | | 12.3 ± 28.68 | 34 ± 6.58 | 6.05 ± 28.7 | 13.5 ± 14.3 |
| 3. | Citronellol | <0.001 | 9.52 ± 1.38 [b] | 3.96 ± 1.27 [ab] | | 1.32 ± 3.51 [ab] | 2.97 ± 0.85 [a] | 25.3 ± 3.51 [c] | 1.7 ± 2.03 [a] |
| 4. | Citronellyl acetate | 0.159 | 4.48 ± 1.21 | 1.36 ± 1 | | | 3.4 ± 0.68 | | 0.27 ± 2.27 |
| 5. | Linalool | 0.308 | 10.3 ± 3.06 | 1.2 ± 2.88 | | 4.02 ± 11.3 | 2.83 ± 2.78 | 4.86 ± 11.3 | 0.83 ± 5.66 |
| 6. | Sabinene | 0.029 | 6.08 ± 4.94 [a] | 27.7 ± 5.05 [b] | | | 7.18 ± 3.79 [a] | 18.1 ± 12.6 [ab] | 17.9 ± 8.94 [ab] |
| 7. | Yield | 0.783 | 1.2 ± 1.39 | 2.96 ± 1.05 | 0.43 ± 2.87 | | 2.55 ± 1.08 | | 3.13 ± 1.65 |

*p*-val.—*p*-value. Different superscript alphabets of means in a row are significant differences based on the least square means at *p* ≤ 0.05.

In terms of plant part predictor, there was a significant result found on most parameters, except caryophyllene. Dry leaves were determined to be the plant part with the highest citronellal content (75.4%), which was not significantly different to fresh leaves (61), leaves (57.1), and twigs (52.2%). Blossoms (partially opened flower) were found to have the lowest citronellal content (6.05%) among several tested parts of the kaffir lime plant. In contrast, the highest citronellol content (25.3%) was found in the blossom of kaffir lime, while the lowest was detected in fruit peels (1.61%). Twigs represented another kaffir lime plant part that was rich in citronellyl acetate (6.2%). The plant part richest in linalool (46.5%) was fresh leaves, while the poorest one (0.81%) was the fruit. In terms of sabinene,

peels, as the richest part, possessed 18.6%, while the leaves were the poorest, with only 4.59%. Concerning yield ($p < 0.001$), the order from high to low among several plant parts was peels > dry leaves > leaves > fruits > twigs (Table 6). In contrast to the results of Tables 6 and 7, the present qualitative meta-analysis revealed that neither the yield or main components were significantly influenced by extraction methods (Table 7).

**Table 6.** Meta-analysis of yield and main components of kaffir lime essential oil in response to plant part (mean ± standard error).

| No. | Parameters | p-Val. | B | DL | F | FL | L | P | T |
|---|---|---|---|---|---|---|---|---|---|
| 1. | Caryophyllene | 0.134 | 3.75 ± 1.23 | | 1.65 ± 0.84 | 0.26 ± 1.23 | 1.62 ± 0.32 | 0.97 ± 0.27 | 1.32 ± 0.57 |
| 2. | Citronellal | <0.001 | 6.05 ± 21.8 [ab] | 75.4 ± 13 [b] | | 61 ± 19.1 [ab] | 57.1 ± 4.52 [b] | 10.6 ± 4.48 [a] | 52.2 ± 10.9 [b] |
| 3. | Citronellol | <0.001 | 25.3 ± 3.11 [c] | | | 11.7 ± 3.11 [b] | 6.23 ± 0.69 [b] | 1.61 ± 0.66 [a] | 7.68 ± 1.53 [b] |
| 4. | Citronellyl acetate | 0.008 | | | 1.74 ± 1.69 [ab] | 1.74 ± 1.98 [ab] | 3.8 ± 0.61 [ab] | 1.74 ± 0.6 [a] | 6.2 ± 1.1 [b] |
| 5. | Linalool | <0.001 | 4.86 ± 8.09 [a] | | 0.81 ± 8.97 [a] | 46.5 ± 6.38 [b] | 3.08 ± 1.92 [a] | 1.64 ± 1.58 [a] | 10.9 ± 4.89 [a] |
| 6. | Sabinene | 0.002 | 18.1 ± 12.2 [ab] | | 6.79 ± 7.2 [ab] | 16.5 ± 7.73 [ab] | 4.59 ± 3.05 [a] | 18.6 ± 2.73 [b] | 5.86 ± 5.09 [ab] |
| 7. | Yield | <0.001 | | 1.72 ± 1.01 [a] | 0.86 ± 1.17 [a] | | 1.63 ± 0.56 [a] | 8.5 ± 0.87 [b] | 0.13 ± 0.92 [a] |

*p*-val.—*p*-value. B—blossom, DL—dry leaves, F—fruit, FL—fresh leaves, L—leaves, P—peels, T—twigs. Different superscript alphabets of means in a row are significant differences based on the least square means at $p \leq 0.05$.

**Table 7.** Meta-analysis of yield and main components of kaffir lime essential oil in response to extraction method (mean ± standard error).

| | Parameters | | Car | Cit | Ctr | CA | Lin | Sab | Yld |
|---|---|---|---|---|---|---|---|---|---|
| | *p*-value | | 0.15 | 0.29 | 0.63 | 0.57 | 0.86 | 0.44 | 0.44 |
| No. | | | | | | Extraction method | | | |
| 1. | | CPE | 1.38 ± 0.72 | 12.1 ± 19.6 | 3.24 ± 3.28 | 1.19 ± 1.75 | 2.36 ± 8.98 | 18.9 ± 7.7 | |
| 2. | | HDSD | 0.4 ± 1.02 | 11.6 ± 25.1 | 1.26 ± 5.42 | | 0.77 ± 6.12 | 37 ± 12.7 | 2.24 ± 2 |
| 3. | | HD | 0.82 ± 0.4 | 34 ± 9.26 | 4.84 ± 1.73 | 2.25 ± 1.07 | 10.1 ± 3.64 | 11.9 ± 4.8 | 2.4 ± 0.93 |
| 4. | | HSD | 0.73 ± 0.88 | 53.5 ± 23.2 | | 1.04 ± 2.49 | 3.85 ± 9.3 | 12 ± 9.4 | |
| 5. | | LNE | | 35 ± 23.2 | 1.29 ± 3.48 | 2.27 ± 2.14 | 1.69 ± 9.3 | 20.2 ± 9.4 | 1.8 ± 2.09 |
| 6. | | SPE | 2.59 ± 0.67 | 7.62 ± 14.8 | 7.83 ± 2.59 | 5.51 ± 2.32 | 4.37 ± 6.19 | 9.96 ± 9.74 | |
| 7. | | SEE | 2.28 ± 0.48 | 41.3 ± 10.4 | 5.17 ± 1.73 | 2.44 ± 1.36 | 2.38 ± 4.9 | 6.85 ± 4.9 | 4.53 ± 1.3 |
| 8. | | SFME | 0.4 ± 1.2 | 17.8 ± 31.3 | 1.24 ± 5.87 | | 0.92 ± 12.3 | | |
| 9. | | SDD | 1.03 ± 0.44 | 47.3 ± 8.84 | 6.52 ± 2.05 | 3.56 ± 1.08 | 2.73 ± 3.11 | 14.4 ± 4.4 | 2.25 ± 0.8 |
| 10. | | SFE | 0.48 ± 1.02 | 1.62 ± 25.1 | 2.17 ± 5.42 | | | | 0.19 ± 1.8 |

Car—caryophyllene, Cit—citronellal, Ctr—citronellol, CA—citronellyl acetate, Lin—linalool, Sab—sabinene, Yld—yield. CPE—cold press extraction, HDSD—hydro diffusion steam distillation, HD—hydro distillation, HSD—hydro steam distillation, LNE—Likens-Nikerson extraction, SPE—solid phase extraction, SEE—solvent extraction, SFME—solvent free microwave extraction, SDD—steam distillation, SFE—supercritical fluid extraction.

In addition to the influence of a single factor, this study also examined the effect of the interaction between factors, as presented in Table 8. The interaction between plant origin and plant part factor was significant on the parameters of citronellyl acetate ($p = 0.048$), sabinene ($p < 0.001$), and yield ($p = 0.031$); whereas, the interaction effect of plant origin and extraction method tended to be significant ($p = 0.067$) only on the citronellol parameter. In addition, the interaction between the plant part and the extraction method tended to be significant in the citronellal ($p = 0.09$) and yield ($p = 0.056$) parameters.

**Table 8.** *p*-value of several interaction among factors: plant origin, plant part, and extraction method on oil yield and main components of kaffir lime essential oils.

| No. | Parameters | PO * PP | PO * EM | PP * EM |
|---|---|---|---|---|
| 1. | Citronellal | 0.712 | 0.663 | 0.09 |
| 2. | Citronellol | 0.314 | 0.067 | 0.305 |
| 3. | Citronellyl acetate | 0.048 * | 0.207 | 0.876 |
| 4. | Linalool | 0.638 | 0.588 | 0.566 |
| 5. | Sabinene | <0.001 * | 0.283 | 0.357 |
| 6. | Yield | 0.031 * | 0.537 | 0.056 |

PO * PP—interaction of plant origin and plant part, PO * EM—interaction of plant origin and extraction method, PP*EM—interaction of plant part and extraction method. The '*' indicated significant interaction on tested parameters at $p \leq 0.05$.

## 4. Discussion

In the last fifty years, numerous single studies have been conducted on kaffir lime essential oils. Due to kaffir lime's importance in various industries, ranging from food and beverages to perfumery and pharmacy, the demand for kaffir lime oils and its raw materials urgently demands support in the form of on-farm and off-farm technology. On-farm technologists have considered the selection of planting materials and growing locations, as well as cultivation practices, whether detrimental or beneficial to the final yield obtained. Kaffir lime planting materials could be in the form of seed, although grafted seedlings were more familiar for commercial cultivation [3]. The growing locations of kaffir lime have not yet been reported on conclusively. Generally, the literature still consists of single studies that examine the influence of climatic and edaphic factors separately. Similarly, cultivation practices were also intensively reported by single studies intending to ensure the optimal raw material harvested, through pruning [27], shading [13], and fertilizing [2,28], while off-farm technologists also considered which method was best suitable to extract the essential oil. Both meta-regression and meta-analysis approaches showed no significant effect of extraction method on oil yield and main components of kaffir lime essential oil. It is likely that all extraction methods were usable, each with their own advantageous characteristics.

### 4.1. Plant Origin

The present meta-analysis elucidated the significant effect of plant origin factor on citronellol, caryophyllene, and sabinene. Citronellol is a natural and important monoterpenoid alcohol compound in kaffir lime with a floral scent [34], which plays a role as a mosquito repellent [86], antifungal [87], and anti-inflammatory agent [88]. USA-derived kaffir lime contained significantly higher citronellol than samples from tropical Asian countries, such as Indonesia, Malaysia, Thailand, Vietnam, and Sri Lanka. The climate difference between the subtropics and tropics was supposed to be the reason for this phenomenon, although the mechanism remains unclear. A test in the same tropical lands concluded that rainfall intensity has a negative correlation to the citronellol content of kaffir lime oil [11]. However, this low citronellol content could be improved by agricultural interventions, such as inorganic and biofertilizer application [89,90].

However, the relative similarity of citronellol content among samples from tropical Asian countries was in contrast to sabinene content. Placing as the second major metabolite in kaffir lime, sabinene, also known as 4(10)-thujene and 4-methylene-1-(1-methyl-ethyl) bicyclo[3.1.0]hexane, is a natural and important monoterpenoid commonly isolated from numerous plant essential oils and popularly used for perfumery [91], due to its fresh and green aroma [34]. Sabinene possesses some pharmaceutical characteristics, such as antifungal [92], antibacterial [93] and anti-inflammatory properties [94]. Even though they are categorized as coming from tropical Asian regions, the sabinene in samples from

Malaysia was higher than in the others. In the case of sabinene, plant origin does not appear to be the single most influencing factor. The interaction between plant origin and plant parts, which is significant ($p < 0.001$) in the sabinene variable, is the main reason for this (Table 8). Most of the single studies obtained from Malaysia used fruit peel samples, and sabinene is the dominant monoterpenoid compound in kaffir lime peel essential oil [22,45].

In addition to monoterpene, kaffir lime essential oil is also composed of certain sesquiterpenes, such as caryophyllene, which could serve as a fragrance and flavor enhancer [95,96], due to its spicy scent [34], in addition to its several pharmaceutical characteristics, such as antioxidant, antibiotic, antimicrobial, and anti-inflammatory properties [88,97]. Although its content values varied (Table 5), caryophyllene could be found in kaffir lime samples from all countries, ranging from subtropical America and tropical Asian countries to tropical countries, such as Mauritius, in East Africa. By employing a meta-analysis method to integrate numerous single studies, it is possible to make a more precise and trustworthy statement than any single study could, i.e., [10,11,98], that caryophyllene is also a main characteristic compound of kaffir lime essential oil.

*4.2. Plant Parts*

In contrast to plant origin, plant parts had no significant effect on the caryophyllene content in kaffir lime oil. Interestingly, citronellol, sabinene, citronellal, linalool, citronellyl acetate, and yield appeared to be significantly affected by plant parts. After combining small single studies, a less popular fact emerged, which is that the flower is the richest plant part in terms of citronellol and sabinene. This information complements the results of previous research, which stated that sabinene is mostly found in the fruit peel [22,45]. The high content of citronellal and sabinene in flowers is thought to be related to the floral fragrance, because these two compounds can produce a floral and fresh aroma [34].

In contrast to the blossom, the leaves are the part of the plant richest in citronellal (Table 6). The meta-analysis carried out to summarize 71 earlier cases on citronellal confirmed that, irrespective of their freshness level, leaves remain the richest part in terms of citronellal, so citronellal mining activities should be balanced with activities to increase kaffir lime leaf production. Citronellal mining is starting to attract attention because it has economic value. As the most intense [99] odor-forming lemony scent [100], citronellal is used as an intermediate component in the synthesis of perfumes, drugs, and basic ingredients for the synthesis of isopulegol, menthol, and citronellol [101–103]. This phytochemical is getting more attention for development in the perfume industry because of its non-toxic nature [104]. Besides its perfumery aspect, this monoterpenoid aldehyde also exhibits some pharmacological properties, such as anti-inflammatory [88], antifungal [87,105], antibacterial [64], and natural mosquito repellent characteristics [106–108].

This meta-study also revealed that differences in the freshness level of the leaves resulted in variation in the linalool content. Linalool is another monoterpenoid alcohol present in kaffir lime essential oils, which has a floral and sweet odor [34] and shows several pharmacological characteristics, namely, antibacterial [109,110], antifungal [87], antidiabetic [111,112], and pest control agent properties [113]. By summarizing 56 single studies on linalool, we have confirmed that, to obtain linalool-rich oils, the leaves' freshness should be maintained. The conclusive reason behind the high variation reported in linalool content was the difference in leaf freshness levels.

Determination of the best plant parts for obtaining citronellyl acetate-rich oil and high oil yield should also consider the plant origin factor, due to the interaction of both factors observed in the present meta-analysis (Table 8). The highest level of citronellyl acetate could be found within the kaffir lime twigs, especially from Indonesia. Citronellyl acetate is the monoterpenol ester converted from citronellol [114], used for perfumery [115] due to its active odor characteristic [116], namely, a fruity and floral scent [34]. Concerning oil yield, the recommendation for the best part to harvest was the fruit peel. However, kaffir lime leaf oil also has economic value, mostly in the Asian market [3]. No less than 21 of 35 single studies making up the meta-analysis database used leaves as an essential oil extraction

material, strengthening the argument that kaffir lime leaves are fragrant cooking spices, due to their richness in essential oils. When combined with the best determination of growing location and proper cultivation techniques, as a basic element of the plant origin factor, the essential oil content of kaffir lime leaves could be increased. It is likely that metabolites within essential oils are highly affected by internal and external factors, including climatic, edaphic, and cultivation practices [117,118]. A positive strong correlation between rainfall intensity and yield of kaffir lime leaf essential oil [11] means strengthening garden irrigation techniques is a key requirement [119–121]. In addition, a positive significant correlation between the oil yield and soil C-organic status [11] implies that manure application is required to boost oil production [122–128].

In contrast to numerous published single studies, the present study integrated more results, i.e., 55 cases of oil yield, 71 cases of citronellal, 53 cases of citronellol, 28 cases of citronellyl acetate, 56 cases of linalool, 46 cases of sabinene, and 46 cases of caryophyllene; thus acquiring greater statistical power to identify discrepancies across studies in a more efficient way. The discrepancy of oil yield and main oil components implied the presence of a knowledge gap among single studies. The first knowledge gap identified in the present study is the high variability of extraction methods; however, there is no standard protocol that is the most effective and efficient. Another gap exists regarding the mechanism of the influence of subtropical and tropical climates that affects the citronellol content of kaffir lime. A hidden fact was revealed, namely, that the flower/blossom is the richest plant part in terms of citronellol and sabinene, complementing the common knowledge that only the leaves and fruit are currently the target in kaffir lime cultivation. To summarize, other interesting facts discovered in this study are as follows: (i) to obtain linalool-rich oils, the freshness of the leaves should be well maintained; (ii) to obtain citronellal-, citronellyl acetate-, and sabinene-rich oils, the extraction should be carried out on leaves, twigs, and fruit peels, respectively; and (iii) oil yield improvement is very challenging because it is not only influenced by plant parts and plant origin, so location-specific and organ-specific cultivation practices need to be formulated. The abovementioned gaps and facts represent interesting directions for future research. Fortunately, we do not need to consider numerous single studies, because a general precise outlook can be found in the present meta-analysis results.

**Supplementary Materials:** The following supporting information can be downloaded at: https://www.mdpi.com/article/10.3390/horticulturae8121132/s1, Table S1: Extracted data from 36 single studies tabulated for meta-analysis study of kaffir lime essential oil variation in the last fifty years.

**Author Contributions:** Conceptualization, R.B.; methodology, R.B. and M.M.S.; software, M.M.S.; validation, R.B.; formal analysis, M.M.S.; investigation, M.M.S.; resources, R.B.; data curation, R.B.; writing—original draft preparation, R.B. and M.M.S.; writing—review and editing, R.B. and M.M.S.; funding acquisition, R.B. All authors have read and agreed to the published version of the manuscript.

**Funding:** This research received no external funding. The APC was fully funded by Universitas Padjadjaran, Indonesia.

**Institutional Review Board Statement:** Not applicable.

**Informed Consent Statement:** Not applicable.

**Data Availability Statement:** Not applicable.

**Acknowledgments:** The authors acknowledge Roedhy Poerwanto (IPB University), Edi Santosa (IPB University), Darda Efendy (IPB University), Andria Agusta (BRIN), and the Faculty of Agriculture, Universitas Padjadjaran, who have increased the authors' interest in kaffir lime research and development. The authors also express their gratitude to the Meta-Analysis in Plant Science (MAPS) Research Group member for technical support during data selection.

**Conflicts of Interest:** The authors declare no conflict of interest.

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
