# Peer review of "Kaffir Lime Essential Oil Variation in the Last Fifty Years: A Meta-Analysis of Plant Origins, Plant Parts and Extraction Methods"

_horticulturae, doi:10.3390/horticulturae8121132_

Round 1

Reviewer 1 Report

This study is interesting , well writen. It needed to be compiled a lot of different results. The reviewer is not a specialist of meta analysis and cannot be critical about the number of samples (sufficient ? ? not too different or similar for instance for the CPE, there are only two studies and about the same part of plant ) . mMay be specify and give an idea of acceptable standard deviation to do a mean ? How do the authors take into account the variation of samples for each case?

The equations 1 and 2 are identical in the M&M , may be precise the specificity for quantitative and qualitative aspect.

In table 2 , higlight the case of linalol , the variation are very important as decribed after some part of plant is the reason . About extraction method caryophyllene is a high  weight  compound compare to the others and the effect of extraction method should be pronounced (when distillation is used) , I'm  not sure it was discussed 

For the discussion, may be specified each part of plant and method to choose for a targert compound.

Reviewer 2 Report

In this manuscript, the authors statistical approach to study the variation of yield and main components of kaffir lime essential oils in response to difference plant origin, plant part and extraction method. I don't see the point of making an article by doing a statistical study of several studies already published and to conclude that the yields of essential oils vary according to the methods of extraction and according to the origin of the plant and the different parts of the plant. I don't see any novelty in that. In general, the research value of this paper seems limited, and the study design is not inadequate.

Reviewer 3 Report

This manuscript titled "Kaffir Lime Essential Oil Variation in the Last Fifty Years: A Meta Analysis of Plant origins, Plant Parts and Extraction Methods" the reason why the review is delayed for a long time, I think the biggest reason may be that the authors cites some equations , and statistical concepts, whose hopes to use this method to integrate some of the research done in the past 50 years. But many reviewers may be unfamiliar with the equation, or even afraid, so they cannot comment. I happen to have a background in engineering, science, agronomy and biomedicine, so this manuscript is not difficult to understand. First of all the reviewer has a few suggestions, as follows

1. The authors have carefully found many references and listed them in the table, but there is a problem that these studies, under the parameters of different origins, different seasons, different times, etc., the authors should put the original data and process of their statistics. For readers to easy read it that original datas must be provided in the section of "Supplementary Materials".

2. The authors themselves mention in the abstract that it is true that the growing location of the source and the plant parts harvested can significantly affect the oil yield. But in this case, the author's research will be less meaningful, unless the authors can use this method to find events that have not been found by previous researchers. Otherwise, it doesn't make much sense to the reader.

3. Page 5, the code ??? of equation (2, this is β1Oij) should be different from equation (1, one is β1Pij). Mathematically, it is impossible to have two the same code, and there will be different parameters, at least a parentheses should be added to differentiate.

4. As to whether the 85 data cited by the authors are applicable to equations (3), (4), (5), the authors should explain more clearly in the discussion section. Now the analysis software is more and more advanced, but whether it is suitable for application here, we do not know. There are great doubts, because for crops, there will be different accompany with growth environments in different years, including climatic factors such as temperature and rainfall. Therefore, it is indeed a difficult task to evaluate with an analytical method, but if the authors thinks that there has been a breakthrough here, even a small difference is a major discovery and worthy publish it. So the author is asked to describe more in the discussion in this manuscript, but it must be logical and please attached the original data for proof.

My suggestion is major revision.

Reviewer 4 Report

The manuscript was well written and it is very original and interesting.  Three main remarks: i) the English should be revised by a native English speaker; ii) the manuscript should be revised in the Introduction because it seems a review work and not a research study; I think that the aim of the study, the positive effects of this research and the main application should be better defined to highlight the important nature of this work.  iii) the conclusions should be written avoiding redundant information already discussed in the other sections and they should be focused only on the main results in relation to the aims of the study. A last suggestion: Results and Discussion sections may be integrated into a single section to avoid repetitions.

Round 2

Reviewer 2 Report

I'am sorry I am still not convinced because despite the authors' comment, I think the manuscript still falls short in terme of novelty and relevance. Furthermore, the manuscript doesn't fit with the special issue's topic.

Reviewer 3 Report

This manuscript has been substantially revised in response to the reviewer's suggestions and should be accepted and published by Horticulturae.